# Thrombocytosis as a Biomarker in Type II, Non-Endometrioid Endometrial Cancer

**DOI:** 10.3390/cancers12092379

**Published:** 2020-08-22

**Authors:** Parker Bussies, Ayi Eta, Andre Pinto, Sophia George, Matthew Schlumbrecht

**Affiliations:** 1University of Miami Miller School of Medicine, Miami, FL 33136, USA; parker.bussies@gmail.com (P.B.); abe38@med.miami.edu (A.E.); 2Department of Pathology, University of Miami Miller School of Medicine, Miami, FL 33136, USA; apinto1@med.miami.edu; 3Sylvester Comprehensive Cancer Center, Miami, FL 33136, USA; sophia.george@med.miami.edu; 4Department of Obstetrics, Gynecology, and Reproductive Science, University of Miami Miller School of Medicine, Miami, FL 33136, USA

**Keywords:** thrombocytosis, endometrial cancer, serous, survival, platelets, race

## Abstract

Thrombocytosis (platelets ≥ 400K) is a common hematologic finding in gynecologic malignancies and associated with worse outcomes. Limited data exist on the prognostic capability of thrombocytosis in women with high-grade endometrial cancer (EC). Our objective was to describe the associations between elevated platelets at diagnosis, clinicopathologic features, and survival outcomes among women with high-grade, non-endometrioid EC. A review of the institutional cancer registry was performed to identify these women treated between 2005 and 2017. Sociodemographic, clinical, and outcomes data were collected. Analyses were performed using chi-square tests, Cox proportional hazards models, and the Kaplan–Meier method. A total of 271 women were included in the analysis. A total of 19.3% of women had thrombocytosis at diagnosis. Thrombocytosis was associated with reduced median overall survival (OS) compared with those not displaying thrombocytosis (29.4 months vs. 60 months, *p* < 0.01). This finding was most pronounced in uterine serous carcinoma (16.4 months with thrombocytosis vs. 34.4 months without, *p* < 0.01). While non-White women had shorter median OS for the whole cohort in the setting of thrombocytosis (29.4 months vs. 39.6 months, *p* < 0.01), among those with uterine serous carcinoma (USC), this finding was reversed, with decreased median OS in White women (22.1 vs. 16.4 months, *p* = 0.01). Thrombocytosis is concluded to have negative associations with OS and patient race.

## 1. Introduction

Endometrial cancer is the most common gynecologic malignancy in the United States [1]. Pathologic features are used to categorize tumors into low- and high-grade subtypes, often described as types I and II, respectively [2]. Type I cancers make up approximately 80% of cases of endometrial cancer, are largely driven by estrogen, and are associated with more than 85% 5 year overall survival. In contrast, type II tumors are typically hormone independent and impart a less than 60% 5 year overall survival [3,4]. Amongst the type II high-grade subtypes are several distinct, though rare, tumor histologies, including uterine serous carcinoma (USC), uterine clear cell carcinoma (CCC), and carcinosarcoma (CS) [5]. The incidence of type II endometrial cancer is increasing [6], for unclear reasons, with racial minorities disproportionately and negatively affected by these specific histologies in both incidence and survival outcomes [7,8].

Thrombocytosis, conventionally defined as platelet counts ≥400,000 (400K), has been a known prognostic biomarker in patients with cancer for more than 50 years [9]. Approximately 40% of patients with thrombocytosis found independent of iron deficiency or benign inflammation have an occult malignancy [9]. Platelets interact with a number of progenitor molecules, including P-selectin, LPA, and β_3_ integrin, to support metastasis and stabilization of tumor microvasculature [10]. High-grade tumors, including high-grade serous ovarian carcinoma and high-grade renal cell carcinoma, have significant associations with thrombocytosis, suggesting that aggressive histologies may be more apt to manipulate this aberrant paracrine signaling pathway, which ultimately results in megakaryocyte overactivation [11,12]. Among patients with ovarian cancer specifically, thrombocytosis has been associated with greater resistance to taxane therapies [13] and is independently prognostic of worse overall survival [11]. Very little robust data exist on thrombocytosis in endometrial cancer. Several small series and meta-analyses have been completed, but these are limited by varying definitions of thrombocytosis, poor histologic characterization, or such small sample sizes that meaningful conclusions are difficult to draw [14,15,16,17]. One study has suggested that platelet levels may be related to histologic grade, but this has not been validated by other investigators [18].

Recognizing the shifting distribution of type II endometrial cancer, deeper investigations into this disease are needed. Our primary objective was to describe patterns of thrombocytosis, and its effects on overall survival, in a cohort of women comprised solely of those with type II, non-endometrioid histologies. Additionally, considering the public health impact type II endometrial cancer has on minority communities, our secondary objective was to evaluate whether thrombocytosis plays a role in differential clinical outcomes by race. Improved characterization of thrombocytosis in this population of women with type II non-endometrioid endometrial cancer may suggest not only its role as a prognostic biomarker, but perhaps a target for therapeutic intervention. 

## 2. Results

### 2.1. Description of Cohort and Characterization of Thrombocytosis

During the study interval, 1964 women were treated for endometrial cancer at our institution. Of these, 273 women with type II, non-endometrioid endometrial cancer met inclusion criteria and were included in the analyses. The cohort demographics are shown in Table 1. The median age at diagnosis was 65 years (range 39–89). There was significant minority representation in the group; 39% were of non-White race, and 67.5% were Hispanic. USC was the most common histology (*n* = 167, 60.7%), followed by CS (*n* = 63, 23%), and CCC (*n* = 45, 16.3%). Nearly 90% of patients had surgery as part of their treatment plans; hysterectomy was a component of all surgical procedures. Sixty-five percent (65%) received chemotherapy. The majority of patients were never smokers (95.9%) and never consumed alcohol (85.3%). At diagnosis, the median CA125 was 38 U/mL, though the range was wide (2 U/mL–12,340 U/mL). More than 50% were diagnosed with advanced stage III–IV disease. ER was positive in 56.1%, and PR was positive in 42.9% of patients. Two-thirds (65.8%) of the tumors had lymphovascular space invasion (LVSI).

The median platelet count for the cohort was 290K (range 93–1079). Approximately one-fifth (19.3%) of the patients had thrombocytosis at diagnosis, with increased frequency in USC (21.8%) cases than in CS (20.1%) and CCC (10.7%) (*p* = 0.71). Mean platelet levels were lowest in those with CS (300k), followed by CCC (330K) and USC (335k) (ANOVA *p* = 0.03). Elevated platelet levels correlated strongly with the presence of LVSI (Spearman’s rho = 0.31, *p* = 0.002), and patients with thrombocytosis experienced significantly higher rates of LVSI than those without thrombocytosis (100% vs. 62.2%, *p* = 0.002). While rising platelet level correlated with rising CA125 (Spearman’s rho = 0.30, *p* = 0.008), there was no significant difference in median platelet level between cases with normal CA125 levels and those with elevated CA125 (315 vs. 282, *p* = 0.24). Thrombocytosis was also more frequently present in advanced-stage disease compared to early-stage disease (6.1% vs 31.3% *p* = 0.001). 

### 2.2. Prognostic Factors Across All Tumor Histologies

The Cox proportional hazards analysis for overall survival is shown in Table 2. In the univariable model, serous histology (HR 2.22 [CI 1.52–3.25]**, *p* < 0.001), advanced-stage disease (HR 2.82 [CI 1.88–4.25], *p* < 0.001), thrombocytosis at diagnosis (HR 2.64 [CI 1.21–3.57], *p* = 0.008), and LVSI (HR 2.53 [CI 1.52–4.22], *p* < 0.001) were all associated with worse outcomes. Only serous histology maintained independent prognostic significance (HR 2.43 [CI 1.22–4.87], *p* = 0.01) upon multivariable regression analysis. 

Associations between thrombocytosis and overall survival were evaluated in a univariable analysis for each histology. Thrombocytosis did not demonstrate significant associations in patients with (CCC) (HR 1.88 [CI 0.21–16.26], *p* = 0.56) or CS (HR 1.52 [CI 0.54–4.32], *p* = 0.43). However, among patients with USC, thrombocytosis was highly associated with worse overall survival (HR 2.39 [CI 1.20–4.75], *p* = 0.01). Given the small sample sizes for CS and CCC, additional multivariable analyses were not performed.

The median unadjusted overall survival for the entire cohort was 55.2 months. Figure 1 shows the survival curves for the entire cohort as stratified by presence or absence of thrombocytosis. Patients with thrombocytosis at diagnosis had a marked reduction in median unadjusted overall survival compared with those who did not (29.4 months vs. 60 months, log-rank *p* = 0.007). Because serous histology was independently associated with worse survival compared to the non-serous histologies, additional analyses by histology and presence/absence of thrombocytosis were performed. The survival curve is demonstrated in Figure 2. Among women with non-serous tumors, median overall survival was 50 months for those with thrombocytosis and 95.9 months for those without. In contrast, women with both serous tumors and thrombocytosis had a median unadjusted overall survival of 16.4 months, while those with serous tumors and no thrombocytosis had a median unadjusted overall survival of 34.4 months (log-rank *p* < 0.001).

### 2.3. Thrombocytosis between Patients of Different Races

Given the known racial disparities in women with type II endometrial cancer, additional analyses with stratification by race were completed. White patients had a median platelet count of 280K, compared to 311K in non-White patients (*p* = 0.07). Despite this, the proportion of women with thrombocytosis did not significantly differ between racial groups (White 19.2% vs. non-White 19.3%, *p* = 0.99). While the presence of thrombocytosis was associated with worse overall survival regardless of race survival of non-White women was less than White women in both thrombocytotic (unadjusted median overall survival 29.4 months vs. 39.6 months) and non-thrombocytotic (unadjusted median overall survival 47.4 months vs. 62.3 months) groups (log-rank *p* = 0.008) (Figure 3a).

### 2.4. Prognostic Factors in Serous Endometrial Cancer

Given that patients with USC represented the largest histologic group within the cohort, and additionally that serous histology exhibited the worst median survival, we further investigated the prognostic value of our clinicopathologic variables in USC cases alone. Thrombocytosis in USC was strongly correlated with presence of LVSI (Spearman’s rho = 0.26, *p* = 0.04), and patients with thrombocytosis experienced a significantly higher prevalence of LVSI than those without thrombocytosis (100% vs. 71%; *p* = 0.04). Thrombocytosis also correlated with rising CA125 (Spearman’s rho = 0.37, *p* = 0.01). There was no difference in proportions of thrombocytosis by race, ethnicity, or positivity for ER or PR.

Table 3 demonstrates the Cox proportional hazards models for patients with USC. In the univariable analysis, advanced stage III–IV disease (HR 1.90 [CI 1.53–2.37], *p* = 2.16), thrombocytosis (HR 2.39 [1.20–4.76], *p* = 0.01) (as noted above), and LVSI (HR 4.53 [HR 1.91–10.75], *p* = 0.001) were all associated with worse overall survival. In the multivariable models, none of these variables independently predicted survival (all *p* > 0.05). Unadjusted median overall survival, however, was significantly reduced in those with thrombocytosis compared to those without thrombocytosis (16.4 vs. 34.4 months, *p* = 0.002).

We further stratified the impact of our independent variables in patients with USC by race. At diagnosis, there were no differences between White and non-White patients in mean CA125 levels, mean platelet levels, presence of LVSI, or presence of estrogen receptor (ER) or progesterone receptor (PR) positivity (all *p* > 0.05). In a univariate analysis, thrombocytosis was associated with worse overall survival in White patients (HR 4.13 [CI 1.46–11.68], *p* = 0.007) but had no association in non-White patients (HR 1.53 [CI 0.59–4.01], *p* = 0.38). While non-White patients exhibited worse survival outcomes than White patients when thrombocytosis was absent (unadjusted median overall survival 30.1 vs. 38.4 months), non-White patients experienced better outcomes than White patients when thrombocytosis was present (22.1 vs. 16.4 months) (*p* = 0.01) (Figure 3b).

## 3. Discussion

While thrombocytosis has been previously studied in patients with endometrial cancer, this study provides the first look at a large cohort of patients with multiple type II histologies, with sub-analyses by race. Serous histology was shown to be a much more lethal disease, with more pronounced effects of thrombocytosis among women with this disease. Additionally, while neither race nor ethnicity appeared to be associated with survival outcomes in the overall cohort, among women with USC, there were surprising disparities by race in the presence of thrombocytosis.

Thrombocytosis as a clinical predictor of worse outcomes has been demonstrated across the gynecologic cancer continuum. In patients with cervical cancer, elevated platelets levels have been associated with worse recurrence free and overall survival, more advanced stage, larger tumors, and greater likelihood of treatment failure [19,20,21]. Among patients with epithelial ovarian cancer, thrombocytosis may be present in up to 62.5% of patients [22]. It has also been associated with advanced stage, worse overall survival, and even an increased risk for postoperative complications such as ileus [23,24]. Similar trends have been reported in studies of thrombocytosis in endometrial cancer patients. Worse survival, increased likelihood of large volume ascites, and advanced-stage disease have all been associated with higher platelet levels [14,25,26]. Our study, utilizing a diverse cohort, validates these reports in a unique population comprised solely of women with high-grade disease. While our study did not find thrombocytosis to be an independent predictor of outcomes, this is not necessarily surprising given how robust both histology and stage are as co-variates.

The mechanisms by which thrombocytosis affects survival have been greatly studied in other disease types, including ovarian cancer. Platelet activation generally has been demonstrated to protect tumor cells from immune surveillance, and enhances metastasis by facilitating endothelial breakdown for transmigration of tumor cells [10,27]. In ovarian cancer, overproduction of both inteleukin-6 (IL-6) and thrombopoetin by the tumors are causative of thrombocytosis, which results in maintenance of tumor microvessel integrity and pericyte support in an autocrine feedback loop [11,28]. In this manner, the tumors actually stimulate their own proliferation. Our finding that LVSI was strongly correlated with thrombocytosis suggests a similar association between platelets and microvascular invasion in high-grade endometrial cancer. Interestingly, in a large study by Stone et al. [11], it was serous histologies that overwhelmingly demonstrated thrombocytosis. As patients with USC had worse overall survival in our cohort, particularly in the setting of higher platelet levels, consideration must be given to a histology-specific driver to this phenomenon.

Thrombocytosis as a clinical entity is important to identify as it may impact cellular responses to therapy. Taxanes are one of the most effective classes of chemotherapy used almost universally in women with advanced and metastatic endometrial cancer. Elevated platelet levels, however, may abrogate the efficacy of those drugs. In a study by Bottsford-Miller et al. utilizing ovarian cancer cell lines co-cultured with platelets, platelets were found to protect against apoptosis and subsequent cell death when treated with docetaxel; in contrast, platelet depletion, and platelet pre-treatment with aspirin, have been associated with improved response to chemotherapy [13]. The same authors noted that upon review of clinical outcomes data, patients with cancers that were refractory to primary chemotherapy (including paclitaxel) all had platelet counts >450K [13]. One retrospective investigation reported similar findings in patients with endometrial cancer [29], albeit in a smaller population that the current study. Animal models for high-grade serous ovarian carcinoma have noted that knockout of IL-6 significantly reduces tumor growth, but when performed in combination with paclitaxel, there was accentuated cell kill [11]. As high levels of serum IL-6 have been reported in USC, the same paraneoplastic pathway for thrombocytosis [30], and thus chemoresistance, may be in action.

Disparities in outcomes in women with endometrial cancer of varying races has been widely reported. Inequitable surgical opportunities, abundant comorbid conditions in Black women, and lack of access to clinical trials have all been cited as contributing to worse outcomes in minority women [31,32,33]. No such data on the differential effects of platelet levels by race have previously been reported; our study is the first. What is interesting, however, is that while non-White women saw survival reductions in the presence of thrombocytosis in the whole cohort, among White women with USC, thrombocytosis actually portended a worse prognosis than non-White women. This finding suggests an underlying difference in the tumor biology unique to USC, perhaps driven by genomic variations in the setting of unique genetic ancestries. Additional study of this finding is warranted, but not in isolation. As marrow tolerance of chemotherapy and inability to dose-escalate has been demonstrated in Black women [34,35,36], other considerations such as drug metabolism, pharmacogenomics, and tumoral epigenetic modifications as mediators of response require attention.

### Limitations

As a retrospective study, our results are limited by the usual biases, including information and misclassification bias. Significant, in-depth analyses are limited by the relatively small number of patients and thus definitive conclusions cannot be drawn from these data. However relative to other published studies on this topic, our population number is robust. In fact, this study represents the largest cohort of high-grade endometrial cancer patients in whom thrombocytosis has been evaluated [17,26]. Additional validation studies should be undertaken to confirm results. The period of study is quite extensive, spanning 12 years. This may introduce temporal bias as chemotherapeutic treatments, opportunities to clinical trials, and access to the public hospital may have varied during this time. Despite this, all patients were treated by the same physician group, so significant variations in practice patterns are unlikely. It is also important to note that malignancy is not the only cause of thrombocytosis. Inflammatory conditions and iron deficiency may also cause elevations in platelets [37], and in our study the data to definitively rule out these conditions was not available. However, inability to rule out alternative causes of thrombocytosis is not unique to our study, and confirmation of malignancy-driven thrombocytosis is rarely performed in the oncologic literature. As a single university-based study, we are biased by the patient population served in our region. However, consistent with the known demographics of South Florida, our study included a significant number of Hispanic (67.5%) and non-White (39%) patients, facilitating additional analyses by unique patient factors, and further generating questions about the biologic role of platelet action in women of different backgrounds.

## 4. Materials and Methods

### 4.1. Standard Protocol Approvals, Registrations, and Patient Consents

A single-institution retrospective review was completed on all women diagnosed and treated for USC, CCC, or CS between 2005 and 2017 at one of three university-affiliated hospitals in Miami, Florida-Sylvester Comprehensive Cancer Center, University of Miami Hospital, and Jackson Memorial Hospital. All patients treated at these facilities during the study interval were identified by the institutional cancer registry, which prospectively collects clinicodemographic, treatment, and outcomes data. All ascertainment activities were approved by the institutional review board (IRB) at the University of Miami (protocol 2015-1022) and adhered to the tenets of the Declaration of Helsinki.

### 4.2. Data Collection

Once eligible patients were identified and their data obtained from the cancer registry, a retrospective chart review was performed to collect the supplemental hematologic data points of interest for this study. Patients were excluded if they received neoadjuvant chemotherapy, had any other synchronous tumor at the time of the diagnosis of endometrial cancer, or if upon chart review they had a type I histology.

Sociodemographic data gathered included age (continuous), body mass index (BMI), smoking status (never/former, current), alcohol use (never/former, current), race (White, non-White [inclusive of Black and other), and ethnicity (Hispanic, non-Hispanic). Pathologic data collected were tumor histology (USC, CCC, CS), cancer stage at diagnosis (I–II, III–IV), estrogen receptor (ER) expression (negative, positive), progesterone receptor (PR) expression (positive, negative), and lymphovascular space invasion (LVSI; positive, negative). Positive hormone expression (PR, ER) was defined as expression in >1% of tumor cells.

Hematologic parameters recorded were CA125 level at diagnosis (<35, ≥35), platelet levels at diagnosis (continuous), and presence or absence of thrombocytosis at diagnosis (<400K, ≥400K). Date of diagnosis (i.e., date of pathologic confirmation) was noted for each patient and used to calculate overall survival, defined as time from date of diagnosis to date of death (all cause). Reports of histology and grade were reviewed by the principal investigator to ensure that the two were consistent. Stage at diagnosis was recorded according to the American Joint Committee on Cancer (AJCC) classifications.

### 4.3. Statistical Analysis

Statistical analyses were conducted using STATA IC 14.2 (StataCorp, College Station, TX, United States). All patients, even those missing specific data points, were included in the analyses to reduce bias. Summary statistics were used to describe the patient cohort. Chi-square tests (or Fisher’s exact, where appropriate) were utilized to examine relationships between categorical variables, and Spearman rank correlation coefficients generated to assess dependence between variables. ANOVA was performed for comparisons of means between groups, and the Kruskal–Wallis test used to make comparisons of non-parametric medians. Univariable and multivariable Cox proportional hazards regression was used to determine associations between clinicopathologic variables and overall survival. Surgery was a covariate was not included in the models given the high prevalence of its utilization throughout the cohort. To avoid inadvertently eliminating potential confounding factors affecting survival, backwards multivariable regression analyses included covariates with *p*-values ≤ 0.05 from the univariable models. Survival data analysis, including curve generation, was performed using the Kaplan–Meier method, with comparisons assessed using the log-rank test. All tests were two sided with significance set at *p* < 0.05, and confidence intervals (CI) were generated at 95% confidence.

## 5. Conclusions

Our study confirms that thrombocytosis is a negative prognostic biomarker in women with type II, non-endometrioid endometrial cancer, but that these negative effects may not be equally distributed histologies, or even among women of different races. These findings suggest that independent of socioeconomic or environmental factors, there may be significant variations in the underlying tumor biology and differential mechanisms by which endometrial cancer propagates. Additional work in larger cohorts should be undertaken to validate these findings, but also seek to explore how, much like in ovarian cancer, platelets drive tumor growth and influence chemotherapy response. Such investigations may ultimately lead to novel and more personalized approaches for treatment in these women for whom few effective options exist.

## Figures and Tables

**Figure 1 cancers-12-02379-f001:**
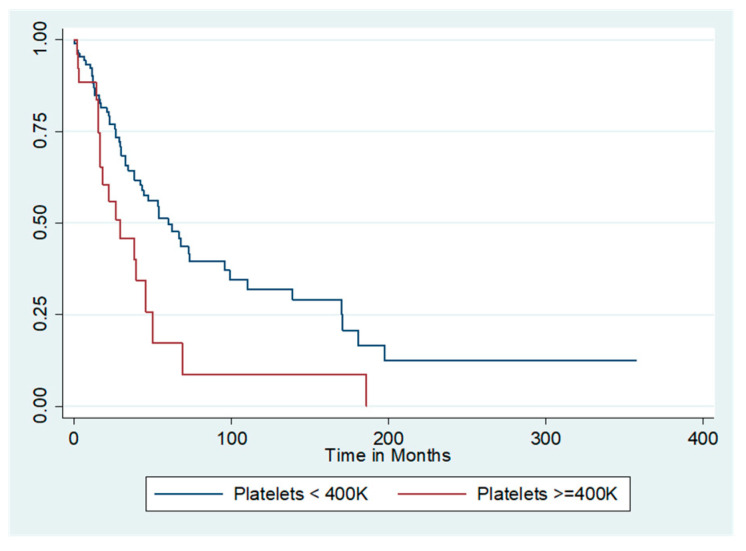
Unadjusted overall survival by thrombocytosis. Median overall survival (OS) was significantly affected by presence of thrombocytosis. Median OS of those with thrombocytosis was 29.4 months compared to 60 months in those without thrombocytosis (log-rank *p* = 0.007).

**Figure 2 cancers-12-02379-f002:**
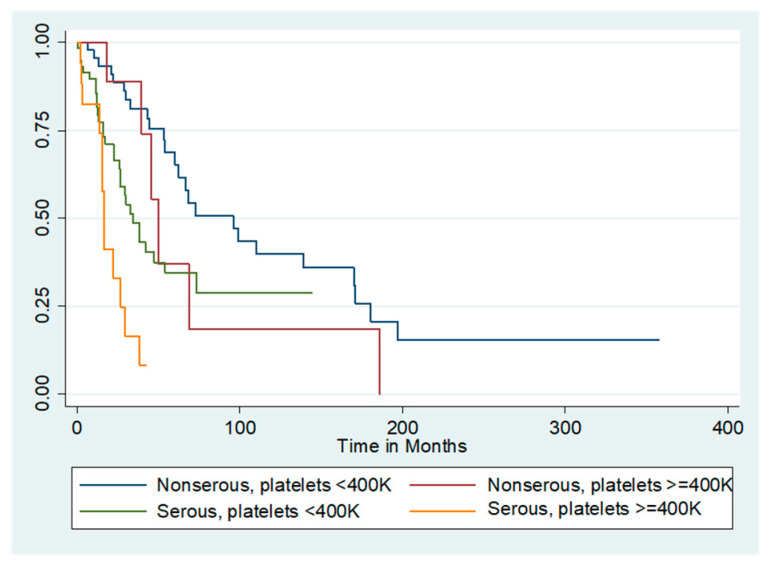
Unadjusted overall survival by thrombocytosis and histology. Median overall survival (OS) of all patients stratified by the thrombocytosis and serous histology. Median OS of non-serous tumors with thrombocytosis was 50 months and was 95.9 months for non-serous tumors without thrombocytosis. In contrast, serous tumors with thrombocytosis had a median OS of 16.4 months, and serous tumors without thrombocytosis had a median OS of 34.4 months (*p* < 0.001).

**Figure 3 cancers-12-02379-f003:**
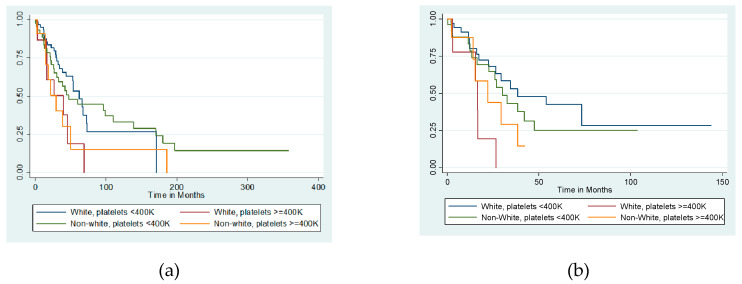
Overall survival by thrombocytosis and race in (**a**) the entire high-grade endometrial cohort (log-rank *p* = 0.008) and (**b**) serous carcinomas only (*p* = 0.01).

**Table 1 cancers-12-02379-t001:** Cohort demographics.

Variable	Number (% or Range)
Age at diagnosis (years)	65 (39–89)
Body mass index (BMI, kg/m^2^) at diagnosis	29.3 (16.8–50.5)
Race	
White	166 (61)
Non-White (Black/other)	106 (39)
Ethnicity	
Hispanic	183 (67.5)
Non-Hispanic	88 (32.5)
Histology	
Serous	167 (60.7)
Clear cell	45 (16.3)
Carcinosarcoma	63 (23)
Tobacco use	
Ever (current/former)	10 (4.1)
Never	235 (95.9)
Alcohol use	
Ever (current/former)	39 (14.7)
Never	204 (85.3)
Surgery	
No	28 (10.3)
Yes	245 (89.7)
Chemotherapy	
No	96 (35.2)
Yes	177 (64.8)
Platelet count	
<400K	109 (80.7)
≥400K	26 (19.3)
CA125 at diagnosis	38 (2–12,340)
Stage at diagnosis	
I–II (early)	110 (47.4)
III–IV (late)	122 (52.6)
Estrogen receptor (ER)	
Positive	64 (56.1)
Negative	50 (43.9)
Progesterone receptor (PR)	
Positive	27 (42.9)
Negative	36 (57.1)
Lymphovascular space invasion (LVSI)	
Present	102 (65.8)
Absent	53 (34.2)

**Table 2 cancers-12-02379-t002:** Univariable Cox proportional hazards modeling for overall survival.

Variable	Univariable			Multivariable		
	HR	CI	*p*-Value	HR	CI	*p*-Value
Age	0.99	0.97–1.01	0.35			
Race						
White (ref)						
Non-White	0.98	0.69–1.42	0.94			
Ethnicity						
Non-Hispanic (ref)						
Hispanic	1.07	0.75–1.54	0.70			
Histology **						
Non-serous (ref)						
Serous	2.22	1.52–3.25	<0.001	2.43	1.22–4.87	0.01
Body mass index (continuous)	0.99	0.96–1.02	0.43			
Smoking						
Never/Former (ref)						
Current	0.49	0.15–1.53	0.22			
Alcohol						
Never/Former (ref)						
Current	0.81	0.48–1.35	0.43			
Stage **						
I–II (ref)						
III–IV	2.82	1.88–4.25	<0.001	1.80	0.89–3.64	0.10
Chemotherapy						
No (ref)						
Yes	1.03	0.73–1.47	0.85			
CA125 Level						
<35 (ref)						
≥35	1.37	0.76–2.47	0.29			
Platelet levels at diagnosis (continuous)	1.002	1.000–1.004	0.007			
Platelet level at diagnosis **						
<400K (ref)						
≥400K	2.64	1.21–3.57	0.008	1.15	0.53–2.51	0.73
P53 expression						
Negative (ref)						
Positive	0.88	0.63–1.24	0.47			
ER expression						
Negative (ref)						
Positive	1.51	0.89–2.53	0.12			
PR expression						
Negative (ref)						
Positive	1.19	0.59–2.39	0.62			
LVSI **						
Negative (ref)						
Positive	2.53	1.52–4.22	<0.001	1.47	0.67–3.24	0.34

** Included in multivariable analysis

**Table 3 cancers-12-02379-t003:** Cox proportional hazards models for overall survival among patients with uterine serous carcinoma.

Variable	Univariable			Multivariable		
	HR	CI	*p*-Value			
Age (continuous)	0.99	0.97–1.02	0.89			
Race						
White (ref)	--					
Non-White	1.09	0.69–1.72	0.70			
Ethnicity						
Non-Hispanic (ref)	--					
Hispanic	0.96	0.62–1.51	0.87			
Body mass index (continuous)	0.96	0.93–1.01	0.099			
Smoking						
Never/Former (ref)	--					
Current	0.56	0.14–2.30	0.43			
Alcohol						
Never/Former (ref)	--					
Current	0.80	0.42–1.52	0.49			
Stage **						
I–II (ref)	--					
III–IV	1.90	1.53–2.37	<0.001	2.16	0.82–5.74	0.12
Chemotherapy						
No (ref)						
Yes	1.01	0.62–1.63	0.97			
CA125 Level						
<35 (ref)	--					
≥35	2.28	0.97–5.33	0.059			
Platelet levels at diagnosis (continuous)						
	1.00	1.00–1.01	0.008			
Platelet level at diagnosis **						
<400K (ref)	--					
≥400K	2.39	1.20–4.76	0.01	1.26	0.53–2.97	0.60
P53 expression						
Negative (ref)	--					
Positive	0.63	0.29–1.42	0.27			
ER expression						
Negative (ref)	--					
Positive	1.24	0.65–2.42	0.51			
PR expression						
Negative (ref)	--					
Positive	0.95	0.41–2.20	0.90			
LVSI **						
Negative (ref)	--					
Positive	4.53	1.91–10.75	0.001	3.81	0.84–17.3	0.08

** Included in multivariable analysis.

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
