# Peer review of "Thrombocytosis as a Biomarker in Type II, Non-Endometrioid Endometrial Cancer"

_cancers, 2020, doi:10.3390/cancers12092379_

Round 1

Reviewer 1 Report

The topic is interesting as high grade EC is related to diminished OS rates and the search for prognostic biomarkers is crucial in this specific EC subtype. The inter-racial analysis is highly valuable and it seems to be limitedly addressed in the literature. Nevertheless, the work is somehow inconclusive and lacks clinical expert opinion.

Major concerns:

  1. The introduction is brief and concise. Perhaps, it needs a more detailed explanation of the molecular implications of the phenomena of thrombocytosis. The authors declare that platelet levels might be related to histological grade but adding some additional information will enrich the introduction.

  1. Methodology is clear and statistical analysis is appropriate. The number of included patients is impressive (N=271) and the collected variables are reasonable. Nevertheless, I miss the collection of some additional variables that have a great impact on survival such as neoadjuvant, adjuvant treatment and pathological complete response. In this line, the collection of progression free survival data could decipher the role of thrombocytosis in tumor progression as authors explain that platelet count could influence tumor response.

  1. Authors show that thrombocytosis is not an independent prognostic factor for OS as observed in the multivariate analysis (table 1). Even though, OS curves between the presence and absence of thrombocytosis are statistically different taking the entire cohort (figure 1). How could these facts be explained? And the same happens analyzing USC group alone. In my opinion, it could be other variables, not previously considered, that might exhibit better prognostic potential in high grade EC.

  1. I find the discussion very enjoyable and well focused. But, what is the final message in the clinical setting? If thrombocytosis is not an independent predictor of OS in high grade EC, which variables have to be taken in account to efficiently predict clinical outcomes? Could authors suggest different management strategies among women of different races?

Minor points

Abstract: define abbreviations of EC and USC as they appear for the first time here.

Author Response

Reviewer 1

Major concerns:

  1. The introduction is brief and concise. Perhaps, it needs a more detailed explanation of the molecular implications of the phenomena of thrombocytosis. The authors declare that platelet levels might be related to histological grade but adding some additional information will enrich the introduction.

We have expanded the introduction, and have referenced the noted association between thrombocytosis and high-grade disease in ovarian and renal malignancies.

Lines 41-48.

  1. Methodology is clear and statistical analysis is appropriate. The number of included patients is impressive (N=271) and the collected variables are reasonable. Nevertheless, I miss the collection of some additional variables that have a great impact on survival such as neoadjuvant, adjuvant treatment and pathological complete response. In this line, the collection of progression free survival data could decipher the role of thrombocytosis in tumor progression as authors explain that platelet count could influence tumor response.

Please forgive a typographic error noted upon review.  The correct N=273 (updated in the text, line 67).  The reviewer points out a number of other clinical variables that may have affected results.  We have clarified the exclusion criteria (neoadjuvant not allowed, lines 262-66), as well as the proportional hazards models to include chemotherapy.  As 90% of patients received a hysterectomy, surgery was not included in the models (line 290-91).

Our cancer registry does not record progression free survival but the reviewer brings up a good point re: response.  We have clarified in the objective that the goal of the endeavor was to evaluate the effects on overall survival (line 58).

  1. Authors show that thrombocytosis is not an independent prognostic factor for OS as observed in the multivariate analysis (table 1). Even though, OS curves between the presence and absence of thrombocytosis are statistically different taking the entire cohort (figure 1). How could these facts be explained? And the same happens analyzing USC group alone. In my opinion, it could be other variables, not previously considered, that might exhibit better prognostic potential in high grade EC.

To further provide more robust data analyses, chemotherapy was included in the proportional hazards models (Tables 2/3), and did not show significant effects on survival relative to the other data points. As 90% of patients received a hysterectomy, surgery was not included in the models (noted in the Methods section line 290-91).

We do highlight in the Limitations section (which appears to have been left out of the initial submission, lines 231-49) that as a small study with low numbers we are limited in the number of variables we could consider and that larger, robust cohorts are necessary to better validate our findings.

  1. I find the discussion very enjoyable and well focused. But, what is the final message in the clinical setting? If thrombocytosis is not an independent predictor of OS in high grade EC, which variables have to be taken in account to efficiently predict clinical outcomes? Could authors suggest different management strategies among women of different races?

We thank the reviewer for this insightful comment. To date there are no treatment guidelines that are racially-driven, but much of that has to do with such little information on the tumor biology of minority women with endometrial cancer. Study has been done on the socioeconomic and access issues producing disparities in survival by race, but very little has been done on differences in genomics, response to treatment, and metastatic potential (significant greater research is required; even the TCGA admits to having too few Black women for genomic profiling to detect low-frequency somatic mutations).  There are some limited data to suggest an inability to dose-escalate and problems with low leukocytes in Black women, implying that perhaps dose-modifications or personalized dosing is necessary, but this is an area that requires much additional study.  We have expanded the last paragraph of the discussion to this end.  It is our hope that hypothesis-generating studies like this one will help drive further understanding of racial disparities in cancer outcomes (lines 224-27).

Minor points

Abstract: define abbreviations of EC and USC as they appear for the first time here.

Abbreviations have been defined (see abstract)

Reviewer 2 Report

I revised the manuscript entitled “Thrombocytosis as a biomarker in high-grade endometrial cancer”.

I was pleased to revise this paper. The authors performed a study aimed to describe patterns of thrombocytosis, and its effects on survival, in patients with high-grade endometrial cancer.

In my honest opinion, the topic is interesting enough to attract the readers’ attention. Nevertheless, there are different points that require to be clarified as reported below.

In general, the Manuscript may benefit from major revisions, as suggested below:

  • I would suggest clarifying the source of data and better reporting on how they were collected.
  • Inclusion and exclusion criteria should be stated. Were patients who underwent neoadjuvant therapy included? Where patients with synchronous ovarian or other cancer included or excluded?
  • Given that platelet count was assessed at diagnosis, was the type of surgery/treatment used to define the study population?
  • Given that high-grade endometrioid endometrial cancer is not included in the study. A general referring to high-grade endometrial cancer is not appropriate starting from the title. I would suggest using a type 2 histology or specifying high-grade non-endometrioid EC.
  • I would suggest reporting the total number of cases diagnosed or managed during the study period.
  • Lines 72-73. In the correlation of Spearman, does thrombocytosis refer to the platelet level?
  • Figures 1 and 2. I would suggest clarifying that these curves are not adjusted for all confounders such as a stage. The reported p-values did not take in to account the stage and other confounders. These figures can be misleading providing only partial information.
  • Table 2. Based on the multivariate analysis, the only factor associated with OS after the adjustment for all other variables is serous histology and this is confirmed with table 3. In this regard, lines 134-136 are unclear as reported. Lines 134-136 is a univariate analysis that simply repeats the association reported in lines 131-132.
  • Table 3. I would suggest revising the title: univariate and multivariate?
  • Overall, results should be revised highlighting more the results of multivariate analysis instead of the numerous univariate analyses that can be a source of misunderstanding.

Author Response

Reviewer 2

  • I would suggest clarifying the source of data and better reporting on how they were collected.

We have updated the first part of the methods section with relevant changes lines 256-266.

  • Inclusion and exclusion criteria should be stated. Were patients who underwent neoadjuvant therapy included? Where patients with synchronous ovarian or other cancer included or excluded?

Thank you for pointing this out.  We have clarified the exclusion criteria (lines 264-266).

  • Given that platelet count was assessed at diagnosis, was the type of surgery/treatment used to define the study population?

These were not used to define the study population; only tumor histology.  We have now included a statement about surgery in the description of the patient demographics (90% received it; all had a hysterectomy, lines 72-74).  Chemotherapy has been described in both the demographics section and was included in the Cox proportional hazards models for overall survival (Tables 2/3)

.

  • Given that high-grade endometrioid endometrial cancer is not included in the study. A general referring to high-grade endometrial cancer is not appropriate starting from the title. I would suggest using a type 2 histology or specifying high-grade non-endometrioid EC.

These changes have been made in the title and throughout the manuscript.

  • I would suggest reporting the total number of cases diagnosed or managed during the study period.

This has been added to the first paragraph of the results section (lines 66-68)

  • Lines 72-73. In the correlation of Spearman, does thrombocytosis refer to the platelet level?

Yes.  This has been clarified to reflect elevated platelet levels (line 82).

  • Figures 1 and 2. I would suggest clarifying that these curves are not adjusted for all confounders such as a stage. The reported p-values did not take in to account the stage and other confounders. These figures can be misleading providing only partial information.

This has been clarified in the figure titles.

  • Table 2. Based on the multivariate analysis, the only factor associated with OS after the adjustment for all other variables is serous histology and this is confirmed with table 3. In this regard, lines 134-136 are unclear as reported. Lines 134-136 is a univariate analysis that simply repeats the association reported in lines 131-132.

Lines 134-136 refer to Table 3, dealing specifically with uterine serous carcinoma.  There are some limitations in multivariable analyses with smaller subsets, especially when the variables considered in the model are related (e.g. LVSI and elevated platelets have a strong correlation).  Our intent was to show that even despite the fact that none of these variables were independently predictive that there may still be an effect.  We have changed these lines to more clearly convey that these are unadjusted survivals.

  • Table 3. I would suggest revising the title: univariate and multivariate?

Title has been changed to simply “Cox proportional hazards models..”

  • Overall, results should be revised highlighting more the results of multivariate analysis instead of the numerous univariate analyses that can be a source of misunderstanding.

We have clarified in a number of places that survival estimates are unadjusted. Even though this is the largest series evaluating thrombocytosis in these tumors, given their rarity and the low numbers, robust statistics to generate hazard ratios in the multivariable analyses is challenging. Multivariable analyses are not feasible for each individual histology.  This has been added directly in the text.

 It appears that our Limitations paragraph was erroneously deleted from the initial submission.  This has been replaced.  We do emphasize in the limitations section that these analyses are limited by the sample size and that larger studies are needed to confirm. As a retrospective review these data are meant to be hypothesis-generating. We have included in the Conclusion section as well that larger cohorts are needed for validation (lines 229-249)

Round 2

Reviewer 1 Report

I thank the authors to submit the revision of the manuscript. In general, I feel that the manuscript has been substantially improved. Introduction has been enriched and additional variables like chemotherapy has been included in the models. I find very valuable to include the limitations of the study in the manuscript although they were erroneally eliminated in the first submission. Almost all concerns have been addressed.

Author Response

Reviewer comments noted and appreciated.

Reviewer 2 Report

I revised the manuscript entitled “Thrombocytosis as a biomarker in type II, non-endometrioid endometrial cancer”.

I was pleased to revise this paper. The authors performed a study aimed to describe patterns of thrombocytosis, and its effects on survival, in patients with high-grade endometrial cancer.

In my honest opinion, the topic is interesting enough to attract the readers’ attention. Moreover, the authors addressed almost all the suggested revision and I appreciated the manuscript improvement. I would suggest, however, to avoid reporting both type 2 and high-grade together as well as endometrioid with low-grade. Grade and histology are two different concepts. If you based the study on type 2 histologies, I would suggest removing the concept of high-grade. In lines 260-261, the authors report: “they had a low-grade (type I) endometrioid adenocarcinoma” This is confusing because I would expect that high-grade endometrioid EC would have been included.

Author Response

Thank you for the comments.  The reviewer is correct in noting that grade and histology are two distinct entities and this is recognized by the authors.  However, we do understand that the fluctuation between 'high-grade' and 'type II' may be unclear.  As such, we have modified the content to use 'type II' instead of 'high-grade' to avoid confusion.

Our secondary aim was to specifically study thrombocytosis in clear cell, USC, and carcinosarcoma as these histologies disproportionately affect racial minorities.  We have added verbiage (line 39) to more specifically highlight this point as it is what was driving the interest in these histologies specifically.

Line 260-261 has been clarified.